# Meteorin-like Protein and Zonulin in Polycystic Ovary Syndrome: Exploring Associations with Obesity, Metabolic Parameters, and Inflammation

**DOI:** 10.3390/biomedicines12010222

**Published:** 2024-01-19

**Authors:** Plamena Kabakchieva, Antoaneta Gateva, Tsvetelina Velikova, Tsvetoslav Georgiev, Kyosuke Yamanishi, Haruki Okamura, Zdravko Kamenov

**Affiliations:** 1Clinic of Internal Medicine, Naval Hospital, Military Medical Academy, 9010 Varna, Bulgaria; plamenakabakchieva@yahoo.com; 2Clinic of Endocrinology, University Hospital “Alexandrovska”, Department of Internal Medicine, Medical Faculty, Medical University—Sofia, 1431 Sofia, Bulgaria; tony_gateva@yahoo.com (A.G.); zkamenov@hotmail.com (Z.K.); 3Medical Faculty, Sofia University “St. Kliment Ohridski”, 1504 Sofia, Bulgaria; tsvelikova@medfac.mu-sofia.bg; 4Clinic of Rheumatology, University Hospital “St. Marina”, First Department of Internal Medicine, Faculty of Medicine, Medical University—Varna, 9010 Varna, Bulgaria; 5Department of Psychoimmunology, Hyogo Medical University, 1-1 Mukogawa, Nishinomiya 663-8501, Hyogo, Japan; k-yama@hyo-med.ac.jp (K.Y.); haruoka@hyo-med.ac.jp (H.O.); 6Department of Psychiatry, Hyogo Medical University, 1-1 Mukogawa, Nishinomiya 663-8501, Hyogo, Japan

**Keywords:** polycystic ovary syndrome, meteorin-like protein, zonulin, obesity, insulin resistance

## Abstract

Objective: Polycystic ovary syndrome (PCOS) is a prevalent hormonal and metabolic disorder, wherein the adipose tissue and gut microbiome have been demonstrated to contribute to its pathogenesis. This study aims to assess the concentrations of the adipokine, meteorin-like protein (Metrnl) and the protein, zonulin, related to intestine permeability, in individuals with PCOS with a particular emphasis on their relationship with obesity, clinical manifestations, hormonal profiles, and metabolic parameters. Methods: A cohort comprising 58 women with PCOS, classified according to the Rotterdam criteria, was enrolled. The study also considered age, body mass index (BMI), and ethnicity-matched controls (*n* = 30). Comprehensive anthropometric and clinical evaluations, hormonal assays, and biochemical analyses were conducted during the follicular phase. Subsequent subgroup analyses were executed within the PCOS cohort based on waist-to-height ratio (WHtR), insulin resistance (IR), and free androgen index (FAI). Serum concentrations of Metrnl and zonulin were quantified via the enzyme-linked immunosorbent assay (ELISA) technique. Results: The Metrnl and zonulin levels exhibited no significant disparity between PCOS patients and controls. Nevertheless, within the entire participant cohort and the PCOS group exclusively, overweight/obese participants demonstrated higher Metrnl concentrations relative to their normal-weight counterparts (*p* < 0.001, *p* = 0.001, respectively). Furthermore, higher Metrnl concentrations were identified in subgroups characterized by high WHtR and IR in comparison to those with low WHtR (*p* = 0.001) and without IR (*p* = 0.001), respectively. A correlation emerged between Metrnl levels and various anthropometric and metabolic parameters, as well as sex-hormone-binding globulin (SHBG) and interleukin-18 (IL-18) within the PCOS group. Multiple linear regression analysis identified HOMA-IR as the sole independent predictor of Metrnl levels. Conclusion: While Metrnl and zonulin levels do not serve as diagnostic indicators of PCOS, elevated Metrnl concentrations exhibited robust associations with proinflammatory and metabolic irregularities within the PCOS population.

## 1. Introduction

Polycystic ovary syndrome (PCOS) stands as the predominant endocrine–metabolic disorder influencing women of reproductive age. Defined by irregular menstrual cycles, hyperandrogenism, and infertility [1], PCOS is accompanied by an array of metabolic perturbations, including insulin resistance (IR), visceral obesity, dyslipidemia, and type 2 diabetes [2], all of which elevate cardiovascular risk [3].

Visceral adipose tissue, recognized as an endocrine entity releasing diverse adipokines [4], constitutes a well-documented risk determinant for the emergence of IR and other metabolic irregularities. Particularly within the context of PCOS, visceral obesity demonstrates a notable role in heightening hyperandrogenism and fueling the onset of correlated metabolic disorders [5]. Our own investigations have confirmed that augmented abdominal adiposity in PCOS women is associated with IR, diminished sex-hormone-binding globulin (SHBG) levels, hyperandrogenism [6], and low-grade inflammation (elevated interleukin-18 (IL-18]) [7]. In light of the available literature and our findings, it can be inferred that visceral obesity holds a pivotal role in the metabolic dysregulation encountered in PCOS. Expanding our comprehension of metabolic anomalies in PCOS demands the examination of additional factors linked to visceral obesity.

IL-18 has emerged as a noteworthy contributor to the metabolic landscape of PCOS. Our previous study illuminated the association between augmented abdominal adiposity in PCOS women and a state of low-grade inflammation characterized by elevated IL-18 levels [7]. This proinflammatory cytokine, known for its involvement in various metabolic disorders, has been implicated in the pathogenesis of insulin resistance, dyslipidemia, and cardiovascular risk [8,9]. Recent years have witnessed explorations into the roles of novel biomarkers, including meteorin-like protein (Metrnl), also known as subfatin, and zonulin, in PCOS development, progression, and interconnected metabolic disturbances.

Metrnl, a newly identified immunoregulatory adipomyokine emanating from white adipocytes, activated monocytes, macrophages [10], and skeletal muscles [11], demonstrates a central role in governing glucose homeostasis and energy metabolism [11,12]. Metrnl accentuates energy expenditure, enhances insulin sensitivity, and prompts gene expression related to thermogenesis and browning of white adipose tissue, along with fostering the production of anti-inflammatory cytokines in mice [11]. This prompts speculation that Metrnl might participate in the pathophysiologic mechanisms of metabolic disorders, including PCOS and type 2 diabetes. Notably, the limited pool of research involving type 2 diabetes patients yields conflicting outcomes [13,14,15,16]. In our understanding, only two studies have evaluated Metrnl levels in PCOS women [17,18], both demonstrating a significant association with metabolic aberrations linked to the syndrome.

In the contemporary scientific landscape, emerging evidence underscores the role of the gut microbiome in the inception of metabolic ailments, as well as in the pathogenesis of PCOS. Zonulin, akin to the Vibrio cholera toxin *Zonula occludens*, has surfaced as a recently identified protein implicated in intestinal permeability regulation and, conceivably, some autoimmune diseases [19]. The nexus between zonulin and metabolic disorders has been recently delineated. Moreno-Navarrete et al. [20] unveiled an association of this biomarker with obesity and insulin resistance, revealing elevated levels in obese and prediabetic patients. However, a handful of studies have dissected this biomarker in PCOS women, with findings being inconclusive [21,22,23], thereby leaving the role of zonulin in PCOS pathogenesis unresolved.

Given the intricate interplay among PCOS, visceral obesity, and insulin resistance, further exploration of these connections is of paramount importance, aiming to unravel the underlying mechanisms and devise more efficacious therapeutic modalities. This study aims to enhance our understanding of the connection between Metrnl, zonulin, and visceral obesity in women with diagnosed PCOS. It also compares these findings with those from healthy individuals, all within the context of insulin resistance and hormonal imbalances.

## 2. Materials and Methods

### 2.1. Patients and Controls

A total of 88 female participants were enlisted in this cross-sectional investigation, comprising 58 individuals diagnosed with PCOS (average age: 25.9 ± 5.2 years) and 30 healthy controls (average age: 27.6 ± 5.2 years). Both groups adhered to inclusion criteria encompassing ages ranging from 18 to 40 years, as well as body mass index (BMI) ranging from 18.5 to 40 kg/m². Patients diagnosed with PCOS were recruited from the endocrinology department of University Hospital Alexandrovska—Sofia, Bulgaria, and their diagnosis followed the Rotterdam criteria [24], guided by the latest international recommendations [25] for the characterization and management of PCOS. Recruitment involved approaching eligible individuals with PCOS during routine clinic visits from May 2019 to January 2020, explaining the study’s purpose, and obtaining informed consent.

Healthy controls constituted women with regular menstrual cycles, devoid of PCOS history or other infertility causatives (apart from male-related factors). These control subjects were meticulously matched based on age and BMI to ensure comparability with the PCOS patients.

Inclusion criteria were applied consistently to all participants; individuals with additional endocrine disorders including thyroid dysfunction, hyperprolactinemia, premature ovarian failure, hypothalamic amenorrhea, congenital adrenal hyperplasia, androgen-secreting tumors, or Cushing’s disease/syndrome were excluded. Those employing combined oral contraceptives, antiandrogens, or insulin sensitizers (such as metformin and thiazolidinediones) within the preceding 3 months were also ineligible. Pregnant or lactating women were not part of the study cohort.

Prior to starting their involvement, each participant provided written informed consent as approved by the Research Ethics Board of the Medical University of Sofia, Bulgaria (Reference number: 16/03.04.2019). Subsequently, the research methodology received approval from the Research Ethics Board of the Medical University of Sofia, Bulgaria (Reference number: 17/02.06.2021).

### 2.2. Anthropometric and Clinical Assessment

A comprehensive medical history covering aspects of menstruation, reproductive experiences, and detrimental behaviors was gathered from all participants. Moreover, a meticulous assessment of anthropometric and clinical parameters was performed by a singular clinician to maintain some bias. This assessment encompassed measurements of height, weight, body mass index (BMI), waist circumference (WC), hip circumference (HC), waist-to-hip ratio (WHR), and waist-to-height ratio (WHtR), along with evaluations for hirsutism, acne, alopecia, and acanthosis nigricans. These evaluation methods were extensively illustrated in our prior investigation [26]. Additionally, an ovarian ultrasound, conducted by an experienced sonographer, was carried out to either confirm or dismiss the presence of polycystic ovarian morphology. For the categorization of obesity, overweight, and normal weight, the World Health Organization (WHO) classifications were adhered to, with obesity defined as BMI ≥ 30 kg/m², overweight defined as BMI ranging from 25 to 29.9 kg/m², and normal weight spanning BMI values of 18.5–24.9 kg/m² [27].

### 2.3. Laboratory Tests

Blood samples were procured from all participants under fasting conditions during the early follicular phase within their spontaneous menstrual cycle (days 3–5) or in response to progesterone-induced menstrual bleeding. These samples were used for measuring parameters such as fasting plasma glucose (FBG), fasting immunoreactive insulin (IRI), luteinizing hormone (LH), follicle-stimulating hormone (FSH), estradiol, testosterone, dehydroepiandrosterone sulfate (DHEAS), 17-OH-progesterone, androstenedione, sex-hormone-binding globulin (SHBG), and IL-18. Additionally, a standard oral glucose tolerance test was administered, and the free androgen index (FAI) as well as the homeostatic model assessment of insulin resistance (HOMA-IR) were calculated following the methodology outlined in our prior research [28].

Fasting serum samples were further collected into serum EDTA tubes and then subjected to centrifugation before being transferred to Eppendorf tubes. These samples were subsequently stored at −80 °C until the evaluation of Metrnl and zonulin levels. The quantification of these two biomarkers was carried out using enzyme-linked immunosorbent assay (ELISA) kits procured from MyBiosource, Southern California, San Diego, USA. The zonulin assay (Cat. No: MBS167049) had a reference range of 0.3 ng/mL to 90 ng/mL, with a kit sensitivity of 0.13 ng/mL. The Metrnl assay (Cat. No: MBS764456) featured a reference range of 78–5000 pg/mL, with a kit sensitivity of <46.9 pg/mL.

### 2.4. Subgroup Analyses

Within the PCOS cohort, we performed subgroup analyses by segregating participants into discrete categories. To be precise, we classified patients into different groups utilizing criteria encompassing WHtR, insulin resistance, and FAI levels, as elaborated in our earlier investigation [7].

### 2.5. Statistical Methods

Statistical analyses were conducted using the IBM SPSS Statistics software, version 21.0, for Windows, based in Chicago, IL, USA. The analytical approach encompassed descriptive statistics, along with parametric and nonparametric tests, in addition to multiple regression analysis. Initially, the distribution’s normality was examined using the Kolmogorov–Smirnov test. In cases where data displayed normal distribution, parametric tests (such as independent samples T-test) were applied, with the outcomes presented as mean differences along with their respective standard deviations (SD). Conversely, when data distribution exhibited skewness, a nonparametric Mann–Whitney U test was employed to compare variables, and the hypotheses are expressed as differences between medians, represented by the interquartile range (IQR). For the analysis of correlation, Pearson’s correlation analysis was employed for normally distributed data, while Spearman’s correlation analysis was chosen for data with non-normal distribution. Multiple regression analyses utilizing the “enter” method for introducing independent variables were performed to discern the primary determinants of the levels of the biomarkers under study among the variables considered. The determination of the sample size was derived from the Metrnl and zonulin measurements in prior investigations [17,21] utilizing the Sample Size Calculator Clinical Calc, available at https://clincalc.com/stats/samplesize.aspx (accessed on 8 November 2023). To achieve a power analysis with a significance level of 0.05 and a power exceeding 80%, it was determined that a minimum of 21 participants per group would be required. Significance was set at *p* < 0.05 for all comparisons, serving as the threshold at which the null hypothesis was invalidated.

## 3. Results

The demographic, clinical, and anthropometric attributes of the study cohort have been previously documented [6,7]. Briefly, the LH/FSH ratio and clinical and biochemical indicators of hyperandrogenism exhibited significant elevation in the patient group compared to those in the healthy cohort, while groups did not differ in demographic, anthropometric, or metabolic features.

### 3.1. Serum Concentrations of Zonulin and Metrnl in PCOS and Healthy Women

Significant disparities in Metrnl and zonulin levels were not observed between the patients and control subjects. The average Metrnl values within the PCOS group and among healthy controls were 365 ± 74.1 pg/mL and 369.7 ± 93.3 pg/mL, respectively. Similarly, the mean zonulin concentrations were 8 [6.9; 11] ng/mL and 9.2 [6.2; 14.6] ng/mL, respectively (*p* > 0.05).

### 3.2. Comparison of Metrnl Levels between Overweight/Obese and Normal-Weight Women in PCOS and Healthy Groups

Metrnl levels were significantly higher in overweight/obese individuals compared to those with normal weight. This difference was observed in both the entire group of participants (with levels of 390.3 ± 85.6 vs. 335.3 ± 61.8 pg/mL, respectively, *p* < 0.001) and specifically in the PCOS subgroup (with levels of 390.5 ± 74.2 vs. 331.3 ± 60.1 pg/mL, respectively, *p* = 0.001). However, there was no such correlation found within the group of healthy individuals (with levels of 390 ± 107 vs. 343 ± 66.7 pg/mL, respectively, *p* = 0.217).

When we compared Metrnl levels in normal-weight women, there was no significant difference between the patient and control groups (331.3 ± 60.2 vs. 343.1 ± 66.7 pg/mL respectively, *p* = 0.528). Similarly, in overweight/obese women, Metrnl levels showed no distinction between PCOS patients and healthy controls (390.5 ± 74.2 vs. 390 ± 107 pg/mL respectively, *p* = 0.546). In summary, elevated Metrnl levels were only noticeable in overweight/obese women compared to those with normal weight, and this pattern was not limited only to the PCOS group (Figure 1).

### 3.3. Subanalyses in the PCOS Group

The cohort of PCOS patients was subsequently divided into subgroups based on waist-to-height ratio (WHtR): WHtR ≥ 0.5 (*n* = 30) and WHtR < 0.5 (*n* = 28). Notably, serum Metrnl levels exhibited a statistically significant elevation in the WHtR ≥ 0.5 subgroup when compared to the WHtR < 0.5 subgroup (395.1 ± 76.1 vs. 332.7 ± 57.3, respectively, *p* = 0.001), as depicted in Figure 2A.

Furthermore, we divided the PCOS group into two based on insulin resistance (IR) status: PCOS with IR (*n* = 25) and PCOS without IR (*n* = 33). The results of this analysis revealed significantly higher Metrnl levels in the PCOS subgroup with IR than in the PCOS subgroup without IR (399.6 ± 74.6 vs. 338.7 ± 62.9, respectively, *p* = 0.001), as shown in Figure 2B.

The third subanalysis demonstrated no significant disparity between the groups defined by FAI: FAI < 5 (*n* = 39) and FAI ≥ 5 (*n* = 19): 354.1 ± 62.5 vs. 387.2 ± 91.4, respectively (*p* > 0.05), as depicted in Figure 2C.

Conversely, the serum zonulin levels demonstrated comparable values across all three subanalyses.

### 3.4. Correlation Analysis of Zonulin and Metrnl across Groups

Metrnl levels correlated significantly with most of the anthropometric and metabolic parameters and SHBG and IL-18 in the PCOS group, as presented in Table 1. In the healthy controls, only one positive correlation was observed between Metrnl and WHtR (r = 0.369; *p* = 0.045). Zonulin did not show a significant correlation with any of the studied parameters in the groups.

### 3.5. Multiple Linear Regression Analysis of Metrnl Levels in the PCOS Group

A multiple linear regression analysis was conducted to identify the main factors among demographic, anthropometric, and metabolic variables that affect Metrnl. In this analysis, Metrnl was treated as the dependent parameter, while age, BMI, and HOMA-IR were employed as independent predictors. The combined factors were able to predict Metrnl levels effectively, as shown by the significant results: F (3; 54) = 8.75, *p* < 0.001, R2 = 0.327. Among these three factors, only HOMA-IR had a statistically significant impact on the predictive model, with a *p*-value of 0.001, as illustrated in Figure 3.

## 4. Discussion

The role of Metrnl as an adipomyokine in metabolic processes has been explored, but its definitive classification as advantageous or unfavorable remains inconclusive. Within diabetic patients, investigations yield conflicting outcomes—Lee et al. [13] and Zheng et al. [14] reported reduced Metrnl serum levels in newly diagnosed type 2 DM patients, while Chung et al. [15] and AlKhairi et al. [16] discovered elevated Metrnl levels in individuals with type 2 diabetes mellitus.

Regarding PCOS, only two studies indicate that Metrnl levels are notably lower in PCOS women than in healthy counterparts [17,18]. Metrnl was inversely correlated with fasting insulin and blood glucose in the PCOS group and positively associated with BMI, adiponectin, and homocysteine levels in controls [17].

Our findings contradict the results of these studies; Metrnl demonstrated no significant divergence between PCOS patients and healthy women. Remarkably, the Metrnl results mirror those from our prior IL-18 study. We observed differences between normal-weight and overweight/obese women. Metrnl levels were markedly elevated in overweight/obese women in comparison to those with normal weight, both across all participants and within the patient group. Conversely, within the control group, no significant difference was observed between healthy women with and without obesity. This parallels AlKhairi et al.’s [16] findings of elevated Metrnl levels among obese participants and discernible differences in adipomyokine levels among obese and type 2 diabetes patients compared to nonobese diabetics. Moreover, our analysis within the healthy group also unveiled no divergence in Metrnl levels between normal-weight and overweight/obese women.

The only difference from our earlier study on IL-18 was that we did not find differences in the adipomyokine levels between normal-weight and overweight/obese women in the healthy group this time. Based on these observations, we may suggest that Metrnl levels in PCOS are somewhat related to body weight, as our analysis demonstrates higher levels in overweight/obese women. However, it is plausible that other factors exert a more substantial influence on its levels.

Preclinical studies on the impact of Metrnl on adipose tissue in mice suggest that its expression by adipocytes heightens insulin sensitivity by stimulating peroxisome proliferator-activated receptor gamma (PPARγ) receptors [29]. Conversely, Metrnl overexpression yields contrary effects, curtailing lipogenesis and suppressing PPARγ expression in human adipocytes, potentially leading to hyperinsulinemia and insulin resistance [12]. Indeed, Löffler et al.’s [12] findings in children diverged from the association observed in mice between Metrnl and amplified thermogenesis and adipose tissue browning. Their comprehensive analysis revealed that this adipomyokine was not tied to the expression of genes associated with adipose tissue browning. Instead, Metrnl hindered adipocyte differentiation, escalated adipocyte hypertrophy, and indirectly triggered hyperinsulinemia [12]. These results contest the earlier postulated anti-inflammatory and antimetabolic properties of Metrnl, linking its confirmed overexpression in obese children’s adipocytes to ensuing clinical observations of BMI-independent hyperinsulinemia and inflammation in human adipose tissue.

In light of these findings, we can conclude that our study unveiled a parallel clinical connection between elevated Metrnl levels, visceral obesity, insulin resistance, hyperinsulinemia, and the proinflammatory factor IL-18. This correlation is supported by the results of our adipomyokine analyses in subgroups stratified by WHtR and the presence or absence of IR. Across both analyses, Metrnl levels remained markedly higher in women with PCOS and visceral obesity than in those with gynoid obesity, and in patients with IR compared to those without.

Correlation analysis corroborated the linkage between visceral obesity and aggravated metabolic profiles with heightened Metrnl levels. Positive correlations emerged with BMI and most anthropometric indicators in the PCOS group, fasting blood glucose and insulin levels, and HOMA-IR. Intriguingly, Metrnl levels exhibited a positive correlation with IL-18, a proven proinflammatory biomarker [30]. Notably, a significant negative correlation manifested only with SHBG, as anticipated. This reflects the established adversarial link between SHBG and IR [31] and the adverse association with visceral obesity, as supported by our analyses.

Within the framework of multiple regression analysis, age, BMI, and HOMA-IR accounted for only 33% of the variance in Metrnl, with solely HOMA-IR emerging as statistically significant in the model. The singular significance of HOMA-IR, a measure of IR, underscores its pivotal role in influencing Metrnl concentrations within women with PCOS. Aligning our findings with existing research [17,18], we can suggest that a deeper understanding of glucoregulatory mechanisms of Metrnl could unveil innovative strategies for treating IR and related metabolic disorders such as PCOS. Moving forward, a more comprehensive exploration of additional factors is warranted to unravel the intricate and unique regulatory mechanisms governing Metrnl in the context of PCOS.

While Zhang et al. [21] reported increased zonulin levels among women with PCOS compared to controls, strongly correlated with insulin resistance, obesity, dyslipidemia, and menstrual disturbance severity in PCOS, and proposed a role for impaired gut permeability in PCOS pathophysiology, we did not yield similar findings. Our zonulin investigations failed to unveil differentiation between the two groups—healthy and PCOS patients—nor did they reveal significant correlations from supplementary analyses. It is however worth noting that, in contrast to our study, the PCOS women in the study of Zhang et al. [21] were more obese and insulin-resistant than the controls. Another study [22] matched PCOS and controls for BMI and metabolic disturbances and observed similar zonulin levels between the groups, comparable to our outcomes. Therefore, we may conclude that zonulin lacks the reliability required as a diagnostic marker for PCOS in women, as observed by Cetin et al. [22]. 

In summary, Metrnl and zonulin could not be used as definitive diagnostic markers for PCOS. Nevertheless, our analysis highlights a distinct connection between heightened Metrnl levels and various markers of generalized and visceral obesity, insulin resistance, and the proinflammatory factor IL-18. These findings suggest that elevated Metrnl levels are linked to proinflammatory and metabolic irregularities in PCOS. Further investigations involving adipomyokines among women with the syndrome are imperative to comprehensively assess their metabolic repercussions.

### Strengths and Limitations

Our research is one of the few studies analyzing Metrnl and zonulin levels among Caucasian PCOS and healthy women. This study addressed clinically relevant aspects by exploring associations between Metrnl levels and hyperandrogenism, insulin resistance, and other metabolic parameters in PCOS patients, which may have implications for tailored PCOS management. Notably, PCOS has many ethnicity disparities, which contribute to the different phenotypes, resulting in heterogeneity of the main metabolic and hyperandrogenic features. The limitations of this study are its small sample size, encompassing only one ethnicity group; the cross-sectional design of the examination; and its single-center nature. Although the study controlled for various demographic, anthropometric, and metabolic factors, there might still be unaccounted confounding variables that could influence the observed relationships. We also acknowledge the complexity of extrapolating results from our proof-of-concept study to clinical practice and the need for caution in interpreting the clinical implications of our findings. Further research, including larger sample sizes among different ethnicity participants and longitudinal studies, could help confirm and expand upon these results.

## 5. Conclusions

Even though Metrnl and zonulin levels cannot be relied upon for diagnosing PCOS, it is important to highlight that elevated Metrnl concentrations exhibited robust associations with proinflammatory and metabolic abnormalities observed in individuals diagnosed with PCOS.

## Figures and Tables

**Figure 1 biomedicines-12-00222-f001:**
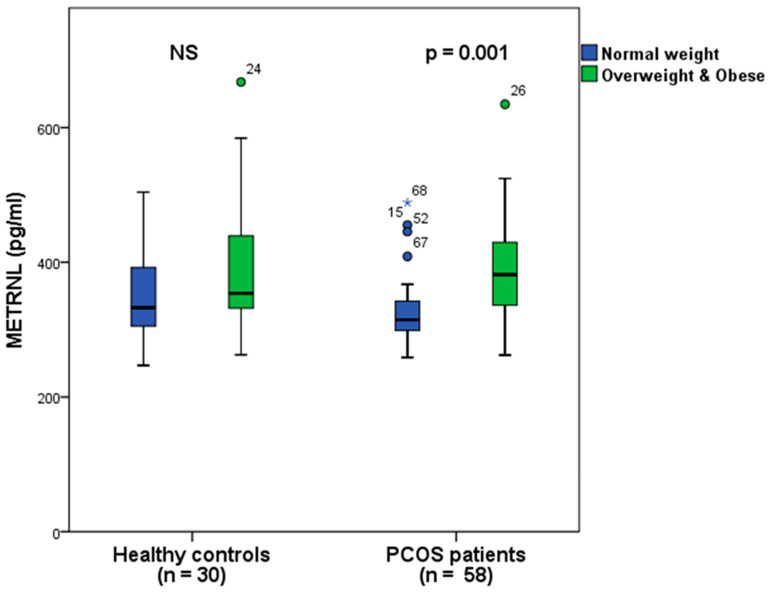
Comparison of Metrnl levels between overweight/obese and normal-weight women in the total study population (*n* = 88) and in both groups, separately. Metrnl levels were significantly greater in overweight/obese women (*n* = 50) versus normal-weight women (*n* = 38) among all 88 participants (*p* < 0.001). In the isolated analysis of the PCOS group, overweight/obese women (*n* = 33) also had higher levels than normal-weight patients (*n* = 25) (*p* = 0.001). In a separate analysis of the healthy group, no similar relationship was identified between overweight/obese (*n* = 17) and normal-weight (*n* = 13) women. No significant difference was observed between patients and controls when comparing only overweight/obese women or normal-weight women in both groups. An asterisk (“*”) indicates an outlier that surpasses three times the interquartile range (IQR), whereas a circle (“o”) signifies an outlier falling between 1.5 IQR and 3 IQR.

**Figure 2 biomedicines-12-00222-f002:**
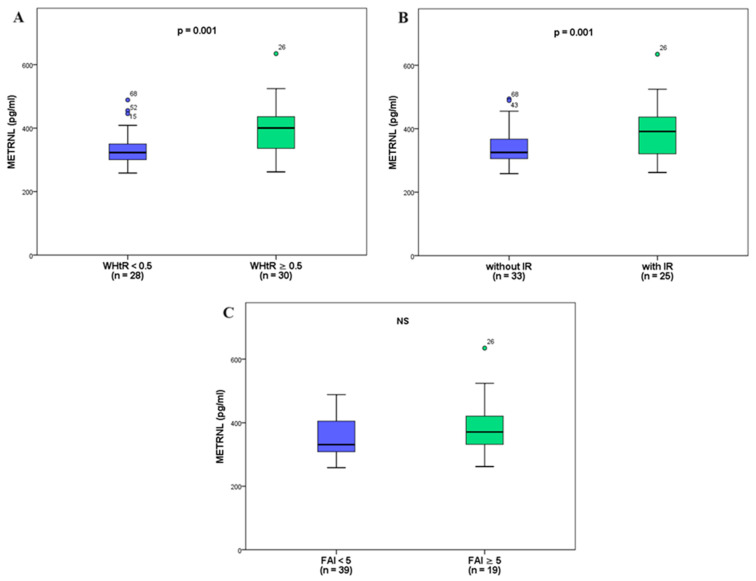
Comparison of Metrnl levels in the three subgroups in PCOS patients: (**A**) comparison between groups with low and high WHtR; (**B**) comparison between groups with/without IR; (**C**) comparison between low and high FAI with no significance. IR—insulin resistance; Metrnl—meteorin-like; WHtR—waist/height ratio; FAI—free androgen index; NS—no significance.

**Figure 3 biomedicines-12-00222-f003:**
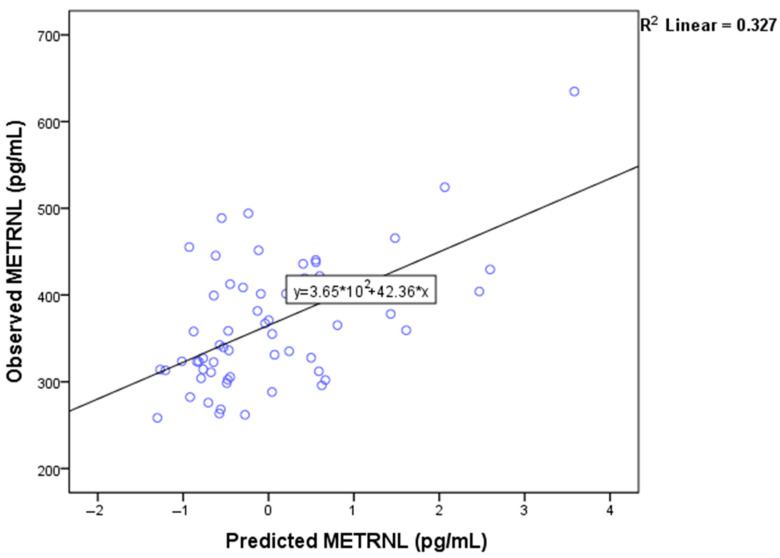
The scatterplot illustrates the relationship between observed and predicted Metrnl values using a linear regression model with Metrnl as the dependent and independent variables—age, BMI, and HOMA-IR. Approximately 33% of the variation in Metrnl could be explained by these three variables, of which only HOMA-IR contributed to statistical significance. BMI—body mass index; HOMA-IR—the homeostatic model for assessment of insulin resistance; Metrnl—meteorin-like protein.

**Table 1 biomedicines-12-00222-t001:** Correlation analysis of Metrnl in the PCOS group.

Parameter	r	*p*
Age (years)	0.169	NS
Weight (kg)	0.452	<0.001
Height (cm)	0.011	NS
BMI (kg/m^2^)	0.368	0.004
WC (cm)	0.423	0.001
HC (cm)	0.445	<0.001
WHR	0.257	NS
WHtR	0.401	0.002
FPG (mmol/L)	0.319	0.015
PG 60′ (mmol/L)	0.178	NS
PG 120′ (mmol/L)	0.148	NS
IRI 0′ (mU/L)	0.391	0.002
IRI 60′ (mU/L)	0.247	NS
IRI 120′ (mU/L)	0.235	NS
HOMA-IR	0.411	0.001
mFG score	0.114	NS
LH (mU/mL)	−0.012	NS
FSH (mU/mL)	−0.137	NS
LH/FSH ratio	0.058	NS
Estradiol (pmol/L)	0.051	NS
Total testosterone (nmol/L)	0.062	NS
DHEAS (µmol/L)	0.045	NS
Androstenedione (ng/mL)	−0.092	NS
FAI	0.255	NS
SHBG (nmol/L)	−0.272	0.039
IL-18 (pg/mL)	0.312	0.017

BMI—body mass index; DHEAS—dehydroepiandrosterone sulfate; FAI—free androgen index; FPG—fasting plasma glucose; FSH—follicle-stimulating hormone; HC—hip circumference; HOMA-IR—homeostatic model for assessment of insulin resistance; IL-18—interleukin-18; IRI—immunoreactive insulin; LH—luteinizing hormone; mFG score—modified Ferriman Gallway scale; PG—plasma glucose; SHBG—sex hormone binding globulin; WC—waist circumference; WHR—waist/hip ratio; WHtR—waist/height ratio. Pearson and Spearman correlation analyses were used in the case of normal and skewed data distribution, respectively.

## Data Availability

Nonconfidential data can be shared on a reasonable basis upon request to the corresponding author.

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
