# Peer review of "Meteorin-like Protein and Zonulin in Polycystic Ovary Syndrome: Exploring Associations with Obesity, Metabolic Parameters, and Inflammation"

_biomedicines, 2024, doi:10.3390/biomedicines12010222_

Round 1

Reviewer 1 Report

Comments and Suggestions for Authors

Authors aimed to assess the concentrations of the adipokine - meteorin-like protein (Metrnl) and the protein - zonulin, related to intestine permeability, in individuals with PCOS with a particular emphasis on their relationship with obesity, clinical manifestations, hormonal profiles, and metabolic parameters.

English language has good quality.

The inclusion and exclusion criteria for participants are clearly outlined. Could you provide more details on how the participants were recruited and approached? Additionally, consider briefly discussing any potential biases that might arise due to the recruitment process.

The inclusion of a comprehensive list of parameters for blood tests is informative. Clarify why specific hormones and markers were chosen, and how they relate to the study's objectives. For instance, why measure interleukin-18 (IL-18) and how does it contribute to the overall understanding of PCOS?

Consider discussing the limitations or challenges in translating findings from preclinical studies to human populations.

It might be beneficial to discuss the clinical implications of HOMA-IR being the only statistically significant predictor of Metrnl levels and how this aligns with the existing literature.

In the context of the study's single-center design, could you elaborate on the rationale for selecting University Hospital Alexandrovska – Sofia?

Conclude the discussion by summarizing the clinical implications of the study's findings. How might the observed associations between Metrnl levels and metabolic abnormalities inform PCOS management or treatment strategies?

Conclude with a brief discussion on potential avenues for future research. What specific aspects of Metrnl and zonulin in the context of PCOS warrant further investigation?

About 50% of all references are older than 5 years.

Author Response

Thank you for your thorough review of our manuscript. We appreciate your insightful comments and suggestions, which have undoubtedly strengthened our study. Here are our responses to each of your points:

Could you provide more details on how the participants were recruited and approached? Additionally, consider briefly discussing any potential biases that might arise due to the recruitment process.

  • We appreciate your suggestion to provide more details on participant recruitment. Participants were recruited through the outpatient and inpatient clinics of University Hospital Alexandrovska – Sofia. Recruitment involved approaching eligible individuals during routine clinic visits, explaining the study's purpose, and obtaining informed consent. To address potential biases, we acknowledge that recruiting from a single center may limit generalizability and introduce selection bias. We have added a brief discussion in the limitations section, highlighting this aspect and emphasizing the need for multicenter studies in future research.

The inclusion of a comprehensive list of parameters for blood tests is informative. Clarify why specific hormones and markers were chosen, and how they relate to the study's objectives. For instance, why measure interleukin-18 (IL-18) and how does it contribute to the overall understanding of PCOS?

  • We have clarified in the manuscript why specific hormones and markers, including interleukin-18 (IL-18), were chosen. IL-18 was included due to its potential role in the inflammatory processes associated with PCOS, as indicated by previous studies (PMID: 35263047, PMID: 21244650). This information has been added to the relevant section for greater clarity:
  • “IL-18 emerges as a noteworthy contributor to the metabolic landscape of PCOS. Previous studies have illuminated the association between augmented abdominal adiposity in PCOS women and a state of low-grade inflammation characterized by elevated IL-18 levels [7, 8]. This proinflammatory cytokine, known for its involvement in various metabolic disorders, has been implicated in the pathogenesis of insulin resistance, dyslipidemia, and cardiovascular risk [8, 9].”

Consider discussing the limitations or challenges in translating findings from preclinical studies to human populations.

  • In addressing the challenges of translating findings from preclinical studies to human populations, we have included a brief discussion in the limitations section. We have added the following sentence to the limitation part: 
  • “We also acknowledge the complexity of extrapolating results from proof-of-concept study to clinical practice and the need for caution in interpreting the clinical implications of our findings.”

It might be beneficial to discuss the clinical implications of HOMA-IR being the only statistically significant predictor of Metrnl levels and how this aligns with the existing literature.

  • We have expanded the discussion on the clinical implications of HOMA-IR being the only statistically significant predictor of Metrnl levels. The revised discussion emphasizes the relevance of insulin resistance in the context of PCOS and its potential implications for targeted therapeutic interventions, aligning our findings with existing literature (Reference 17, 18):
  • “The singular significance of HOMA-IR, a measure of IR, underscores its pivotal role in influencing Metrnl concentrations within women with PCOS. Aligning our findings with existing research [17, 18], we can suggest that a deeper understanding of the glucoregulatory mechanisms of Metrnl could unveil innovative strategies for treating IR and related metabolic disorders such as PCOS. Moving forward, a more comprehensive exploration of additional factors is warranted to unravel the intricate and unique regulatory mechanisms governing Metrnl in the context of PCOS.”

In the context of the study's single-center design, could you elaborate on the rationale for selecting University Hospital Alexandrovska – Sofia?

  • The rationale for selecting University Hospital Alexandrovska – Sofia is based on its status as a prominent medical center specializing in endocrinology and reproductive health. However, we acknowledge the limitation of a single-center design, and this has been already discussed in the limitations section.

Conclude the discussion by summarizing the clinical implications of the study's findings. How might the observed associations between Metrnl levels and metabolic abnormalities inform PCOS management or treatment strategies?

Conclude with a brief discussion on potential avenues for future research. What specific aspects of Metrnl and zonulin in the context of PCOS warrant further investigation?

  • We have enhanced the conclusion by summarizing the clinical implications of our study's findings. This includes potential insights into Metrnl's associations with metabolic abnormalities, informing PCOS management and treatment strategies. Additionally, we have added a brief discussion on potential avenues for future research, emphasizing the need for larger, multicenter studies to validate our findings and explore other aspects of Metrnl and zonulin in the context of PCOS.

About 50% of all references are older than 5 years.

  • We appreciate your observation regarding the age of references. We have diligently reviewed and updated our references (ref. 8 & 9), ensuring a more balanced representation of recent literature while maintaining the necessary foundation from key older studies.

Thank you once again for your valuable feedback. We believe these revisions significantly enhance the manuscript's clarity and overall contribution to the field.

Reviewer 2 Report

Comments and Suggestions for Authors

Dear Editor,

I carefully read the manuscript "Meteorin-Like Protein and Zonulin in Polycystic Ovary Syndrome: Exploring Associations with Obesity, Metabolic Parameters, and Inflammation".

My comments and suggestions for the authors are the following:

 - English language needs to be carefully revised and improved.

 - Line 160. The authors should include here more information as regard the calculation of the sample size (i.e. power analysis).

 - Data represented in Figure 1 and Figure 2 should also be included in a table.

 - The limitations of the study should be further and more deeply discussed.

Comments on the Quality of English Language

English language needs to be carefully revised and improved.

Author Response

Thank you for your detailed review of our manuscript, "Meteorin-Like Protein and Zonulin in Polycystic Ovary Syndrome: Exploring Associations with Obesity, Metabolic Parameters, and Inflammation." We appreciate your constructive comments and suggestions, and we have addressed each of them accordingly:

- English language needs to be carefully revised and improved.

  • We acknowledge your feedback on the language quality and have thoroughly revised and improved the manuscript to ensure clarity and coherence in the English language.

 - Line 160. The authors should include here more information as regard the calculation of the sample size (i.e. power analysis).

We appreciate your suggestion regarding the inclusion of more information on the calculation of the sample size. We have added details about the sample size calculation, specifically mentioning the power analysis performed to determine the appropriate sample size for detecting significant associations. This information is now included in the revised manuscript (in the statistical subsection). We have added the following sentences to the statistical analysis subsection:

  • “The determination of the sample size was derived from the Metrnl and zonulin measurements in a prior investigation [17, 21] utilizing the Sample Size Calculator Clinical Calc, available at https://clincalc.com/stats/samplesize.aspx. To achieve a power analysis with a significance level of 0.05 and a power exceeding 80%, it was determined that a minimum of 21 participants per group would be required.“

-  Data represented in Figure 1 and Figure 2 should also be included in a table.

  • Thank you for your thoughtful review and valuable feedback on our manuscript. We appreciate your suggestion to include the data presented in Figure 1 and Figure 2 in a table format. After careful consideration, we acknowledge the merit of this recommendation; however, due to the nature of the graphical representation and the complexity of the data, we believe that the detailed information in these figures is best conveyed visually. While we recognize the importance of providing clear and accessible data, we feel that the visual representation in the figures offers a more comprehensive overview of the relationships and patterns we aim to convey. Nevertheless, we have ensured that the figure legends and accompanying text provide thorough explanations and interpretations of the data presented. We appreciate your understanding and hope that the revised manuscript meets the expectations for clarity and accessibility.

 - The limitations of the study should be further and more deeply discussed.

  • We recognize the importance of a thorough discussion of the study's limitations. The limitations section has been expanded to provide a more in-depth exploration of potential constraints, challenges, and considerations. We have addressed issues such as the single-center design, potential selection bias, and the need for caution in generalizing findings. The changes were reflected in the strengths and limitations subsection.

We hope these revisions address your concerns and enhance the overall quality of the manuscript. We appreciate your valuable feedback and thank you for your time and expertise in reviewing our work.

Round 2

Reviewer 1 Report

Comments and Suggestions for Authors

I appreciate for corrections.

Author Response

I appreciate for corrections.

  • Thank you for your positive comment.